# Participation and returns from informal service-oriented non-farm enterprises: Evidence from a survey of Nigerian households

**Ikechukwu Darlington Nwaka**[1]*, **Okechukwu Lawrence Emeagwali**[2]

**1** Department of Marketing, Business Economics and Law, Alberta School of Business, University of Alberta, Edmonton, Canada, **2** Department of Business Management, Girne American University, Kyrenia, North Cyprus, via Mersin 10, Istanbul, Turkey

* ike.nwaka@ualberta.ca, ikenwaka@gmail.com

**Data Availability Statement:** All relevant data are within the paper and its Supporting information files.

## Abstract

We investigate the factors that influence the selection and productivity of informal service-oriented family enterprises in Nigeria. Using nationally representative micro-data from the Nigerian General Household Survey (2010–2015), we employed random-effect probit and selectivity-adjusted regression models to estimate and analyze the results. The findings reveal that the location of informal Non-Farm Household Enterprises (NFHEs)–whether home-based or non-home-based—significantly impacts the wholesale, retail, personal, and consultancy service sectors operated by informal NFHEs. This impact remains significant even after accounting for variations in individuals, households, or locational characteristics. Furthermore, when considering selectivity in the earnings equation, we found that home-based informal enterprises exhibit lower productivity compared to non-home-based enterprises, a difference that varies across sectors. Overall, factors such as the gender of business owners, educational levels, geopolitical zones, infrastructure, and business characteristics play a crucial role in determining the locational and productivity disparities among service-oriented enterprises in Nigeria. Key recommendations stemming from this study include addressing gender-based segregation and economic disparities, prioritising financial inclusion for small business development, bridging infrastructure gaps, and implementing policies that acknowledge and bolster the informal sector.

## 1. Introduction

How do household characteristics influence the engagement and efficiency of informal Non-Farm Household Enterprises (NFHEs)? This inquiry delves into the complex dynamics of the informal economy in Nigeria and other sub-Saharan African (SSA) nations. According to the Nigerian General Household Survey's panel data from 2010 to 2015, 94% of household enterprises were informal, defined as those not officially registered with the government. Of this, 48% (refer to Table 1) of employment was generated by informal businesses operating outside

**Funding:** The author(s) received no specific funding for this work.

**Competing interests:** The authors have declared that no competing interests exist.

the household's premises (referred to as non-Home-Based Enterprises (non-HBEs) hereafter), while the remaining 52% operated within the household environment (known as Home-Based Enterprises (HBEs) hereafter).

The existing literature highlights three key factors that differentiate Home-Based Enterprises (HBEs) from non-HBEs. Firstly, HBEs often incur constrained initial business costs, typically financed through the owner's personal savings and without the need for expenses related to business space, as opposed to non-HBEs that necessitate such payments and other financial obligations [1]. Secondly, the time-use cost for HBEs is significantly lower than that for non-HBEs, resulting in more flexible outcomes in value creation between household and labor market productions. This reduced time-use cost implies relative ease, including lower transport costs from home to work, facilitating a better balance between economic and household production, particularly for women [2,3]. Thirdly, HBEs offer what [4] describes as fungible consumption patterns within the household, allowing for the swift conversion of available business resources into alternative uses. These distinctions suggest that factors influencing the choices of service-oriented enterprises may vary across locations, contributing to productivity differences between HBEs and non-HBEs.

Descriptive data from the World Bank [5] highlights the increasing importance of Nigeria's service sector within its economy. Between 2000 and 2015, the contribution of the agricultural sector to GDP declined from 26% to 20%, while the service sector more than doubled its contribution, reaching approximately 59% in 2015. This shift is attributed to structural reforms and diversification policies, establishing the service sector as a potential force in Nigeria's economy [6]. The growing significance of the service sector has given impetus to the emergence of informal enterprises (see Table 1), particularly in wholesale and retail trade, constituting approximately 53% of employment during the study period. However, the growing service sector has seen non-Home-Based Enterprises (non-HBEs) outperform Home-Based Enterprises (HBEs) in average labour productivity, with non-HBEs recording 28,872 Naira/month (approximately 150 US Dollars) compared to HBEs' 19,620 Naira/month (approximately 102 US Dollars—based on 2015 official exchange rate of 192.44 per the US Dollars). This productivity gap, influenced by factors like workforce size, reflects global concerns about the efficiency of informal employment. The challenges in the informal sector could compromise Nigeria's drive for human and economic development, as highlighted by international research, such as [7], showing non-HBEs to be approximately 47% more productive than HBEs.

**Table 1. Informal nonfarm household enterprise employments in nigeria—Percentage share and productivity differences (2010–2015).**

|  | HBE | Non-HBE | OVERALL |
|---|---|---|---|
| Formal | 3 | 8 | 6 |
| Informal | 97 | 92 | 94 |
| **Informal NFHEs** |  |  |  |
| Overall Percentage Employment Share | 51.53 | 48.47 | 100 |
| **Sectoral Employment Shares** |  |  |  |
| Wholesale and Retail Trade | 49.70 | 58.55 | 53.99 |
| Consultancy and Personal Services | 19.21 | 23.72 | 21.40 |
| Manufacturing | 31.09 | 17.73 | 24.62 |
| **Productivity** |  |  |  |
| Average Labour Productivity/Month in Naira | 19,630 | 28,872 | 24,092 |
| Median Labour Productivity/Month in Naira | 3,657 | 8,916 | 5,627 |

Source: Authors' Computation using GHS Cross-Sectional Panel Data (2010–2015).

Despite their significant contribution to employment and welfare, little is known about the determinants of service operation in Nigeria. This study aims to address two fundamental questions:

- What are the determinants of participation and productivity within Nigerian informal service-oriented NFHEs, specifically wholesale/retail trade versus personal/consultancy services?

- Do the determinants of productivity in such enterprises differ based on location (HBE and Non-HBEs)?

Utilizing the General Household Survey Cross-Sectional Panel Data (2010–2015), this study finds that the location of informal NFHEs (HBE or non-HBE) significantly impacts wholesale/retail and personal service-oriented enterprises, even after accounting for individual, household, or geopolitical differences. Controlling for selectivity in the earning equation reveals that home-based informal enterprises are less productive compared to non-home-based, varying across wholesale/retail and personal service-oriented enterprises.

This paper is structured as follows: Section 2 reviews the context of the informal sector as well as the nature of informal service-oriented enterprises in Nigeria, while section 3 details data and variable definitions. Sections 4 and 5 present the research methods and empirical findings, and finally, section 6 concludes and offers policy recommendations.

## 2. Context and informal home-based employments

### 2.1 The context: Informal sectors

Over the years, terminology used to describe the informal sector has evolved, with various schools of thought contributing to this discourse. [8] extensively reviewed conceptualisations of informality, introducing terms such as cash-in-hand, undeclared, hidden, black, shadow, and underground, among others. Despite seeming synonymous, three main terminologies —'informal', 'undeclared', and 'shadow' economy—have gained prominence in labour literature [8–17]. Despite different terms and definitions, they all refer to economic activities outside the state's purview, lacking tax obligations and government support.

This study adopts the term 'informal sector' from [17] due to its comprehensive nature, aligning with the nature of our dependent data (household data). However, within this sector's categorical activities as delineated by [17], our focus falls under 'paid informal work'. Our conceptualisation of informal enterprises aligns with the Fifteenth International Conference of Labour Statisticians' (15th ICLS) definition from 1993 [18], defining informal enterprises based on the following criteria:

- Unincorporated, owned, and managed by a household member.

- Market-oriented, producing goods and services for sale.

- Employing a specified number of workers or own account workers.

- Non-agricultural and not officially registered with the government.

Informal employment is often seen as a survival activity for the very poor [19]. Diverse perspectives exist regarding the causes, composition, and nature of the informal economy. Dualists, according to ILO [20], perceive it as unregulated, comprising small, family-oriented businesses. Structuralists [21] view it as heterogeneous due to actions by formal capitalist firms responding to structural changes. Legalists [22] attribute its existence to legal and bureaucratic formalisation processes, while voluntarists [23] argue that it exists by choice. The informal

sector's role in national economies remains contentious, lacking consensus [24]. One prevalent theory suggests that it leads to loss of tax revenues for the state [25], but [10] argues that it's more complex than portrayed.

In the context of Non-Farm Household Enterprises (NFHEs), a significant portion is considered service oriented. [7] highlight that informal non-farm enterprises are vital sources of livelihood and economic development, generating around 70% of employment, as noted in the World Bank's review of household enterprises in sub-Saharan Africa [5]. Non-farm enterprises encompass a range of activities beyond agriculture, including agribusiness, wholesale and retail trade, construction, utilities, commerce, tourism, and other services [26–28].

## 2.2 Home-based and non-home-based service oriented enterprises

In the literature, the decision to situate informal enterprises either at home (HBEs) or away from home premises (non-HBEs) arises from three distinct factors. Firstly, the cost-minimization hypothesis ties closely to HBEs, as their business start-up finances often stem from owners' personal savings, leveraging the available space within home premises. In contrast, non-HBEs require more substantial capital for start-ups, partly due to the expenses associated with business spaces in various industrial locations or distant open areas [29,30]. Secondly, HBEs minimize time-use costs considerably more than non-HBEs, resulting in flexible outcomes for value creation between household and labour market productions. This reduced time-use cost includes lower transport expenses from home to work, facilitating the balancing of economic and household production, particularly for women [31,32]. Notably, household production involves child-care activities while simultaneously managing HBEs. Literature also suggests potential productivity efficiency reductions in HBEs due to inherent managerial lapses [33,34] which contrasts with the typically higher productivity in on-site non-HBEs.

Thirdly, HBEs offer what [4] terms as 'fungible consumption patterns' within households, indicating the swift conversion of available business resources into alternative uses. Earnings from informal sales easily transition into domestic consumption or business reinvestment, enhancing family members' living standards. However, this flexibility might also lead to potential losses due to negative spillover effects, contributing to location-based heterogeneity (heterogeneous concept is loosely defined to imply different characterization of the informal nonfarm enterprises and differences in factors affecting its performance) between HBEs and non-HBEs. Under the concept of 'extended fungibility,' [4] suggests that HBEs can readily convert household resources into domestic or economic consumption, unlike non-HBEs. These differences imply that factors influencing service-oriented enterprise choices may vary by location, resulting in productivity disparities between HBEs and non-HBEs. Generally, businesses oriented towards wholesale and retail trade necessitate open spaces in public or on-site locations to cater to service demands. Therefore, it is argued that non-HBEs are more oriented towards wholesale and retail businesses compared to HBEs [4,29,30]. Moreover, the composition of outputs from wholesale and retail businesses requires direct market proximity, differing from personal and consultancy services that might rely on a large residential clientele base.

## 2.3 Nature of informal service-oriented household enterprises in Nigeria

The Nigerian General Household Survey Panel Data provides extensive information on non-farm family businesses across 5,000 households and their members. Surprisingly, over 95% of these households engage in at least one non-farm enterprise or income-generating activity. Consequently, the analysis will focus on survey reports concerning non-farm (non-agricultural based) enterprises.

**Table 2. Weighted distribution of informal service oriented enterprises by geopolitical zones, region and owner's gender (2010–2015).**

| | Home Based | | | | Non-Home Based | | | |
|---|---|---|---|---|---|---|---|---|
| | Wholesale & Retail | | Personal & Consultancy | | Wholesale & Retail | | Personal & Consultancy | |
| | Number (in Millions) | % | Number (in Millions) | % | Number (in Millions) | % | Number (in Millions) | % |
| **Zones** | | | | | | | | |
| North-Central | 2.701 | 11.19 | 0.777 | 8.74 | 3.954 | 15.01 | 1.196 | 10.49 |
| North-East | 3.329 | 13.80 | 1.529 | 17.21 | 2.110 | 8.01 | 0.654 | 5.73 |
| North-West | 5.886 | 24.39 | 2.579 | 29.02 | 3.409 | 12.94 | 1.327 | 11.64 |
| South-East | 1.388 | 5.75 | 0.450 | 5.06 | 4.405 | 16.73 | 1.591 | 13.95 |
| South-South | 3.902 | 16.17 | 0.939 | 10.56 | 4.440 | 16.86 | 1.294 | 11.35 |
| South-West | 6.922 | 28.69 | 2.614 | 29.41 | 8.017 | 30.44 | 5.341 | 46.84 |
| **Region** | | | | | | | | |
| Urban | 10.543 | 43.70 | 3.451 | 38.83 | 12.671 | 48.11 | 6.811 | 59.73 |
| Rural | 13.584 | 56.30 | 5.437 | 61.17 | 13.664 | 51.89 | 4.593 | 40.27 |
| **Gender** | | | | | | | | |
| Female | 19.157 | 79.40 | 5.369 | 60.41 | 16.145 | 61.31 | 3.000 | 26.30 |
| Male | 4.970 | 20.60 | 3.519 | 39.59 | 10.190 | 38.69 | 8.404 | 73.70 |

Source: Authors' Computation using the GHS Cross-Sectional Panel Data (2010–2015).

Table 2 displays the distribution of informal service-oriented enterprises categorized by geopolitical zones, regions, and gender between 2010 and 2015. It's notable from the table that a higher concentration of both HBEs and non-HBEs is evident in the South-West region. This concentration could be attributed to the high population density in Lagos, offering extensive commercial and economic opportunities. Interestingly, HBEs, particularly those offering personal and consultancy services, exhibit a higher concentration in rural areas compared to urban settings. Conversely, non-HBEs focusing on personal and consultancy services are predominantly concentrated in urban areas.

Furthermore, an analysis of HBEs and Non-HBEs according to owners' gender reveals distinct patterns. Female enterprise owners tend to operate more home-based businesses, but they participate in non-home-based enterprises primarily within the wholesale and retail trade-oriented sector. In contrast, male owners show higher representation in non-HBEs, especially in the personal and consultancy-oriented enterprises. This observation indicates potential gender-related differences in the operation and preference for HBEs and non-HBEs within Nigeria.

An essential question to consider is whether post-secondary education plays a decisive role in operating an HBE or Non-HBE. Upon reviewing Fig 1, it becomes evident that primary and secondary education significantly influence wholesale and retail-oriented enterprises (both HBEs and Non-HBEs). However, for personal and consultancy-oriented enterprises, it's apparent that secondary education emerges as the primary determining factor.

Another crucial question to address is: do revenues from HBEs and Non-HBEs differ across service-oriented enterprises and gender? Table 3 details the sales revenue of HBEs and Non-HBEs categorized by enterprise activity and gender. The data illustrates that the mean sales revenue of HBEs tends to be lower than that of Non-HBEs in wholesale and retail enterprises. In contrast, revenues from consultancy-oriented enterprises in HBEs are notably higher than those in Non-HBEs on average. Examining the median revenue gap(calculated at the median such ((Non-HBE- HBE)/Non-HBE)*100 or Revenues from Male-owned—Female-owned/

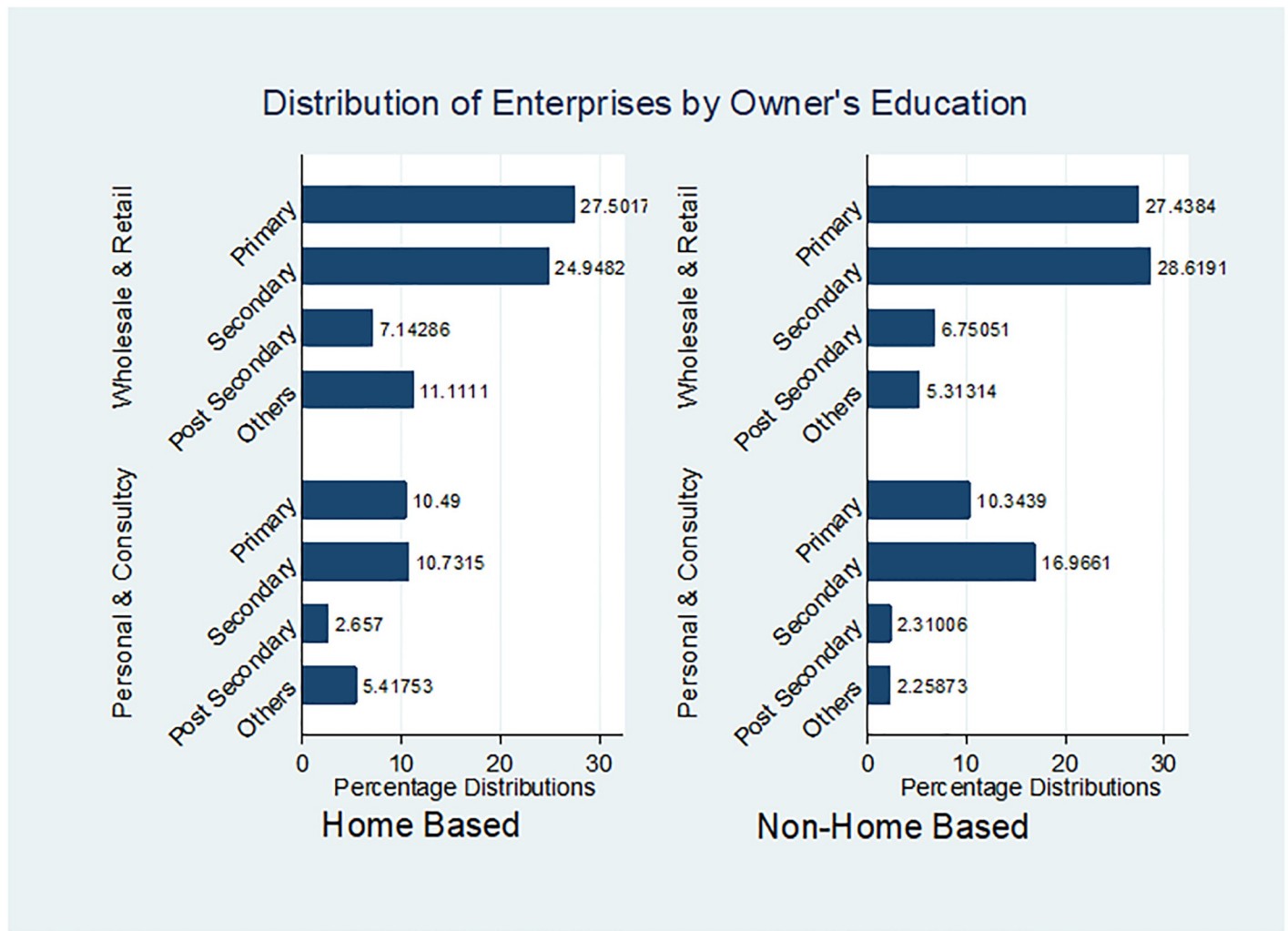

**Fig 1. Distribution of enterprises by owner's education.** Source: Authors' Computation using GHS Cross-Sectional Panel Data (2010–2015).

**Table 3. Mean and median sales revenue by (2010 and 2015), by location, service sector and gender.**

|  | Wholesale and Retail Trade (WRT) | Consultancy and Personal (CP) | Revenue Gap (Median %) | Male-Owned | Female-Owned | Gender-Ownership Gap |
|---|---|---|---|---|---|---|
| **HBEs** |  |  |  |  |  |  |
| Mean | 26,263 | 141,465 | 25.4 | 119,373 | 13,805 | 46.7 |
| Median | 8,039 | 6,000 |  | 10288 | 5,486 |  |
| **Non-HBEs** |  |  |  |  |  |  |
| Mean | 66,784 | 72,096 | 15.2 | 92,621 | 31,049 | 12.5 |
| Median | 17,685 | 15,000 |  | 17,146 | 15,000 |  |
| **Median Revenue Gap (%)** | 54.5 | 60 |  | 39.9 | 63.43 |  |

Source: Authors' Computation using GHS Cross-Sectional Panel Data (2010–2015).

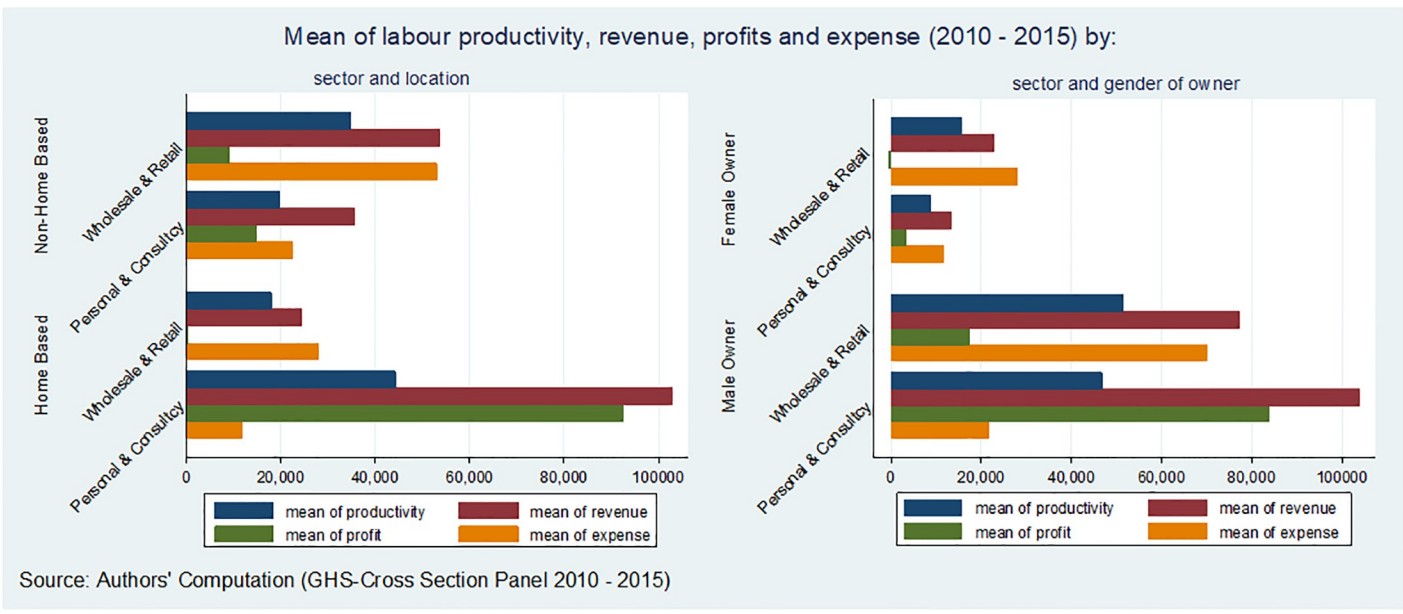

**Fig 2. Average labour productivity, revenue, profit, and expense (2010–2015).** Source: Authors' Computation using GHS Cross-Sectional Panel Data (2010–2015).

Male-owned *100) reveals that Non-HBEs earn approximately 55% and 60% more in wholesale and retail trade enterprises, respectively, than HBEs. Notably, within HBEs alone, revenue from wholesale and retail-oriented enterprises surpasses that of consultancy and personal enterprises by 25%. Similarly, in Non-HBEs, wholesale and retail trade enterprises yield 15% more revenue than consultancy and personal services. Moreover, median revenue from male-owned enterprises tends to be larger in Non-HBEs, reflecting a revenue gap of 40%. This trend remains consistent for female-owned enterprises, where Non-HBEs generate about 63% more revenue than HBEs.

When comparing gender differences in revenue earnings, there is a wider gender revenue gap in HBEs (47%) compared to non-HBEs (13%). This significant revenue gap in HBEs might indicate an increasing female role within households beyond solely operating the enterprise. Additionally, Fig 2 depicts the average labor productivity, profits, and expenses for both HBEs and Non-HBEs, considering gender. Interestingly, mean profits are highest in consultancy and personal services for HBEs compared to non-HBEs, and notably, business profits are substantially higher in male-owned enterprises.

This analysis underscores potential heterogeneity within informal service-oriented enterprises, stemming partly from concentration and productivity disparities. The study further aims to account for other plausible factors contributing to this heterogeneity, taking into account household and enterprise characteristics. Subsequent sections will delve into this aspect in more detail.

## 3. Data and variable definitions

### 3.1 Data

The analysis utilized the GHS panel data from the Nigerian Bureau of Statistics (NBS), encompassing three waves (2010/2011—wave 1, 2012/2013—wave 2, and 2014/2015—wave 3). This dataset is nationally representative, comprising 5,000 households selected from various regions

across the country. The questionnaires utilized in this study include household and community questionnaires from the post-planting and post-harvest sections. The household section captures demographic information, education, employment details, and non-farm enterprises. In contrast, the community counterpart records access to basic facilities and infrastructure.

The conceptualization of the informal sector in this study adheres to the conventional definition, defining informal household enterprises as those not officially registered with the Corporate Affairs Commission (CAC), aligning with the 15th ICLS definition. Informal enterprises, for the purpose of this research, are identified as those responding "NO" to the survey question "Is this enterprise/business officially registered with the government?" Moreover, the term 'service-oriented enterprise' in this paper encompasses the activities outlined in Appendix A, broadly categorized as Wholesale, Retail and Trade, or Consultancy and Personal Services.

The sample for analysis comprises 3,255 households engaged in non-farm enterprises. Overall, the sample consists of 5,557 HBEs and 5,377 non-HBEs from the three waves. Tables 4 and 5 present the definition of variables and summary statistics, detailing various determinants of service-oriented enterprises and productivities for both HBEs and non-HBEs. Specifically, Table 5 presents mean differences across wholesale retail trade and personal/consultancy services to assess whether the characteristics vary across sectors and business locations.

Within the HBEs, significant mean differences are observed in age groups (<25 and >55), education (primary), gender, geopolitical zones (North-Central, North-East, West, or South-South), rural locations, electricity connection, and productivity across wholesale retail trade and personal/consultancy-oriented enterprises. These distinct variables may account for potential sources of heterogeneity, indicating that factors influencing the choices between HBE wholesale retail trade and personal/consultancy-oriented types differ within each sector.

Comparatively, characteristics observed in non-HBEs exhibit even more heterogeneity, showcasing significant mean differences across wholesale retail trade and personal/consultancy-oriented businesses. This disparity between HBEs and non-HBEs underscores the heterogeneous nature of labour demand and supply in the Nigerian economy, considering labour demographics, household, infrastructure, and business characteristics integrated into the probit models and earning equations as outlined below.

## 3.2 Dependent variables

This study employs multiple dependent variables. Initially, a binary variable (1 for wholesale and retail trade, 0 for consultancy and personal service) will be used in the probit model. Subsequently, the estimation of value added per worker (productivity) will be conducted through pooled OLS and random effects estimates.

## 3.3 Independent variables: Labour and demographic characteristics

Labour and demographic factors aim to capture the heterogeneity among informal household owners, considering geopolitical zones and regions. Age categories will delineate the impact of different age brackets on informal business ownership and productivity. Studies suggest that individuals under 25 years in SSA are less likely to own informal enterprises compared to older age groups [7].

Educational levels serve to represent the effects of human capital endowment. Notably, a substantial proportion of both HBEs and non-HBEs fall within the Middle Age range (26 to 45 years) or below the post-secondary education level, as indicated in Table 5. The categorical variable of educational level is used instead of the continuous years of schooling in order to

**Table 4. Variable definitions.**

| Variables | Variable Description |
|---|---|
| *Dependent Variables* | |
| **Probit Model** | |
| Informal Wholesale/Retail and Trade Services | = 1 if is wholesale/retail and trade oriented which is not officially registered with the government and = 0 if such businesses operates as nonfarm Consultancy and personal services oriented that is not officially registered with the government. |
| **Earnings Equation** | |
| Log of prod | Value added per worker = Total Revenue/No of workers |
| *Independent Variables* | |
| **Labour and Demographic** | |
| **Characteristics** | |
| *Age Groups (Base < = 25)* | Age of group of the family business owner (Base category = = 0 if between 15–25 years |
| 26–35 | = 1 if between 26–35 years |
| 36–45 | = 1 if between 36–45 years |
| 46–55 | = 1 if between 46–55 years |
| > 55 | = 1 if above 55 years |
| *Education (Base = Others)* | Level of education of the business owner relative to individuals categorized (adult educated, Quoranic education or did not complete primary education) |
| Primary | = 1 if owner has primary education |
| Secondary | = 1 if owner completed secondary education |
| Post-Secondary | = 1 if owner has completed a post-secondary education |
| Gender_owner | = 1 if male owner, 0 otherwise |
| Zones (Base = North-Cent) | Geopolitical zones (Base = North-Central) |
| North-East | = 1 if located in the North-East |
| North-West | = 1 if located in the North-West |
| South-East | = 1 if located in the South-East |
| South-South | = 1 if located in the South-South |
| South-West | = 1 if located in the South-West |
| Rural | = 1 if located in the rural areas, 0 otherwise. |
| **Household Characteristics** | |
| HH-size | Number of household members |
| Own_home* | = 1 if owned residential homes, 0 otherwise |
| Paid-HH* | = 1 if Business has a paid member of the household |
| Unpaid-HH* | = 1 if Business has an unpaid member of the household |
| **Infrastructure** | |
| Electric | = 1 if improvement in electricity connection in household, 0 otherwise |
| Transport | = 1 if improvement in electricity facility in the community, 0 otherwise |
| Micro-finance | = 1 if micro-finance institution is in the community, 0 otherwise |
| **Business Characteristics** | |
| Profit* | Profit = Net revenues = Total sales revenue—Total costs |
| Turnover | Sales-to-Total-Assets Ratio = Sales/Total Assets |

Note:

* are excluded for the earnings equation (exclusion restriction). Note: Informality as used here reflects businesses that are not officially registered with the government at the time of the survey.

**Table 5. Summary statistics for the HBEs and non-HBEs informal service oriented enterprises.**

| | HBEs | | | | | | Non-HBEs | | | | | |
| | Wholesale & Retail Trade | | Consultancy & Personal | | Mean Differences | | Wholesale & Retail Trade | | Personal & Consultancy | | Mean Difference | |
| Variables | Mean | SD | Mean | SD | t-stat | p-value | Mean | SD | Mean | SD | t-stat | p-value |
|---|---|---|---|---|---|---|---|---|---|---|---|---|
| **Ages Categories** | | | | | | | | | | | | |
| (Base Group< = 25) | 0.105 | 0.306 | 0.129 | 0.335 | 3.112** | 0.002 | 0.070 | 0.255 | 0.125 | 0.331 | 6.166** | 0.000 |
| 26–35 | 0.248 | 0.432 | 0.261 | 0.439 | 0.845 | 0.398 | 0.227 | 0.419 | 0.259 | 0.438 | 2.045** | 0.041 |
| 36–45 | 0.240 | 0.427 | 0.263 | 0.441 | 0.462 | 0.644 | 0.273 | 0.446 | 0.267 | 0.443 | -1.878 | 0.060 |
| 46–55 | 0.164 | 0.370 | 0.167 | 0.373 | 0.243 | 0.808 | 0.201 | 0.401 | 0.174 | 0.380 | -1.525 | 0.127 |
| > 55 | 0.244 | 0.429 | 0.180 | 0.385 | -4.062** | 0.000 | 0.228 | 0.419 | 0.174 | 0.380 | -2.925** | 0.004 |
| **Owner's Education** | | | | | | | | | | | | |
| Primary | 0.389 | 0.488 | 0.358 | 0.480 | 2.098** | 0.036 | 0.403 | 0.491 | 0.324 | 0.468 | -3.520** | 0.000 |
| Secondary | 0.353 | 0.478 | 0.366 | 0.482 | 1.539 | 0.061 | 0.420 | 0.494 | 0.532 | 0.499 | 5.296** | 0.000 |
| Post-Secondary | 0.101 | 0.301 | 0.0907 | 0.287 | -1.226 | 0.220 | 0.0991 | 0.299 | 0.0725 | 0.259 | -2.350** | 0.019 |
| Others = Base | 0.157 | 0.364 | 0.185 | 0.388 | 1.710 | 0.085 | 0.0780 | 0.268 | 0.0709 | 0.257 | -0.845 | 0.398 |
| Male_owner | 0.227 | 0.419 | 0.418 | 0.493 | 13.515** | 0.000 | 0.395 | 0.489 | 0.757 | 0.429 | 22.888** | 0.000 |
| **Geopolitical zones** | | | | | | | | | | | | |
| Zones (Base = North-Cent) | 0.147 | 0.354 | 0.120 | 0.326 | -2.801** | 0.005 | 0.206 | 0.404 | 0.148 | 0.355 | -4.202** | 0.000 |
| North-East | 0.180 | 0.384 | 0.209 | 0.407 | 4.128** | 0.000 | 0.0903 | 0.287 | 0.0829 | 0.276 | -0.571 | 0.568 |
| North-West | 0.258 | 0.437 | 0.305 | 0.460 | 2.499** | 0.013 | 0.131 | 0.338 | 0.130 | 0.336 | -0.529 | 0.577 |
| South-East | 0.0622 | 0.242 | 0.0591 | 0.236 | -0.539 | 0.589 | 0.184 | 0.387 | 0.161 | 0.368 | -2.064** | 0.039 |
| South-South | 0.161 | 0.367 | 0.102 | 0.303 | -3.879 | 0.000 | 0.168 | 0.374 | 0.137 | 0.344 | -2.305** | 0.021 |
| South-West | 0.192 | 0.394 | 0.205 | 0.404 | -0.639 | 0.523 | 0.221 | 0.415 | 0.342 | 0.474 | 8.551** | 0.000 |
| Rural | 0.653 | 0.476 | 0.685 | 0.465 | 2.227** | 0.026 | 0.606 | 0.489 | 0.521 | 0.500 | -5.287** | 0.000 |
| HH-size | 8.139 | 4.073 | 8.207 | 3.900 | 0.659 | 0.510 | 7.308 | 3.374 | 6.810 | 3.368 | -3.532** | 0.000 |
| Own-Home | 0.759 | 0.428 | 0.753 | 0.431 | 0.872 | 0.383 | 0.658 | 0.475 | 0.582 | 0.493 | -4.262** | 0.000 |
| Paid_hh | 0.320 | 0.467 | 0.337 | 0.473 | 0.488 | 0.625 | 0.321 | 0.467 | 0.348 | 0.477 | 1.381 | 0.168 |
| Unpaid_hh | 0.308 | 0.462 | 0.284 | 0.451 | -1.733 | 0.083 | 0.320 | 0.467 | 0.274 | 0.446 | -3.048** | 0.002 |
| electr | 0.559 | 0.497 | 0.522 | 0.500 | -2.242** | 0.025 | 0.606 | 0.489 | 0.640 | 0.480 | 2.301** | 0.021 |
| com_trns | 0.134 | 0.341 | 0.133 | 0.339 | -0.089 | 0.929 | 0.145 | 0.352 | 0.127 | 0.333 | -0.919 | 0.358 |
| Microfin. | 0.104 | 0.305 | 0.088 | 0.284 | -1.684 | 0.092 | 0.811 | 0.273 | 0.994 | 0.300 | 2.028** | 0.042 |
| Turnover | 1.058 | 3.428 | 0.859 | 2.395 | -1.408 | 0.159 | 2.262 | 34.91 | 2.357 | 30.73 | -0.219 | 0.826 |
| Profit | 485.6 | 214,943 | 92,594 | 2.845e+06 | 1.701 | 0.089 | 9,237 | 802,360 | 14,760 | 86,797 | 0.054 | 0.957 |
| Log Productivity | 8.523 | 1.457 | 8.272 | 1.485 | -5.277** | 0.000 | 9.308 | 1.372 | 9.047 | 1.320 | -5.265** | 0.000 |
| Observations | 2772 | | 1049 | | | | 3176 | | 1228 | | | |

Note:

**$p < .05$; SD = Standard Deviation; t-stat = t-statistic.

control for potential measurement errors [2,35,36]. Geopolitical zones and regions help control for potential effects on labour heterogeneity, aligning with the findings of [2].

Gender dynamics in business ownership within HBEs and non-HBEs exhibit distinct patterns in developing countries, where female household owners are more inclined towards HBEs than males. The gender of business owners reflects the likelihood of male or female participation in service-oriented enterprises within both HBEs and non-HBEs, including associated productivity disparities. Table 5 illustrates a larger proportion of male owners in non-HBEs compared to HBEs.

**Household characteristics.**   Households vary in attributes and sizes, potentially impacting business operations and productivity. Variables such as larger household sizes, homeownership, and the presence of paid or unpaid household members will be included in the probability and earnings models. Notably, homeownership can serve as a source of financial capital, influencing the probability of operating informal businesses [36].

**Infrastructure and business characteristics.**   The presence of electricity, transportation facilities, and access to microfinance institutions can stimulate enterprise operations. Infrastructure deficiencies have been identified as inhibiting factors for non-farm enterprises in developing countries [37,38]. Additionally, the turnover ratio, indicating an enterprise's efficiency in generating revenue relative to its assets, will be considered. Table 5 indicates higher turnover in non-HBEs across industries compared to HBEs.

**Exclusion restriction variables.**   Variables such as home-ownership, paid and unpaid household members, and enterprise profits will serve as exclusion restriction variables, featured in both the probit model and earnings equation.

## 4. Empirical model

Central to the research questions, the initial phase of the study will involve modelling the probability of operating either a wholesale/retail trade or consultancy/personal service-oriented informal enterprise. This will be achieved using a pooled panel and random effects probit regression framework. To ensure an unbiased estimate of the productivity model, the predicted probabilities derived from the probit model will be integrated into the subsequent stage of the analysis. This step is crucial to identify and account for any potential selectivity bias within the primary models of interest [39,40].

This study focuses on informal non-farm enterprises in Nigeria, categorizing them into two groups: home-based (HBEs) and non-home-based (non-HBEs). These enterprises are assumed to primarily engage in either wholesale and retail services or personal/consultancy services, as determined by data categorizations. However, our research exclusively concentrates on service-oriented enterprises, deliberately excluding those oriented towards manufacturing. This deliberate exclusion enables a meticulous analysis of the distinctive characteristics and attributes unique to service-oriented enterprises. The productivity model (referred to as the earnings equation henceforth) for each mode is determined as follows:

$$lnProd_{sit} = X_{sit}\beta_s + \mu_{sit} \ for \ each \ HBEs \ and \ non-HBEs \tag{1}$$

where S = 1 for wholesale and retail service-oriented sector or S = 1 for consultancy and personal service-oriented sectors. $lnProd_{si}$ is the value added per worker (average labour productivity) for a given enterprise $i$ in time "t" (such that t = 2010, 2012 and 2015) for each service-oriented enterprise $S$ The vector of broad categorization of variables describing enterprise characteristics, such as the labour and demographics, household, infrastructure, and business characteristics are represented by X (broad categories are listed and defined in (Table 4).

). The β is a vector capturing parameter to be estimated, while $\mu$ is a vector of random disturbance term with a zero mean. We posit that the sectoral choices of non-farm informal employees are conditioned by their utility derived from engaging in either S (wholesale and retail trade or personal/consultancy services oriented) [41].

Thus,

$$O_i = MaxV_{si} \tag{2}$$

where $V$ is the employee's/business owner's unobserved utility derivable from each employment mode, and $O_i$ is the associated benefit employees derived from S. By transforming (2) as

a linear function of employees' or enterprise observed characteristics and unobserved heterogeneity among the various service-oriented workers and enterprises, we obtain:

$$V_{si} = U_i \gamma_s - \varepsilon_{si} \qquad (3)$$

Where, $\gamma$ represents a vector of parameters, $U_i$ is a vector of variables explaining sectoral choices while $\varepsilon$ represents the random disturbance term having a zero mean. The probability of each enterprise being observed as a (wholesale/retail trade = 1 versus consultancy and personal services = 0) can be written as:

$$P[(V_i = S = 1)] = \Pr[S > 0] = \emptyset(y_i \delta + \omega_i) \qquad (4)$$

where, $y_i$ are vectors of exogenous observed characteristics as shown in Table 5, $\delta$ is the parameter vectors due to y, $\omega_i$ is the disturbance term while $\emptyset$ is the univariate probit function.

Eq (4) therefore, represents the pooled probit model which does not account for other time invariant unobserved heterogeneity in the panel structure. Thus, the study further implements the random effects probit model by controlling for the time-invariant unobservable in the disturbance term, such that $\omega_{sit} = \theta_s + \epsilon_{sit}$. Hence, $\theta_s$ captures the effects time invariant unobservable characteristics such as differences in informal service-oriented classification or differences in owner's motivation in running any of the HBEs or non-HBEs [22]. Similarly, $\epsilon_{sit}$ captures other unobservables. The $\theta_s$ and $\epsilon_{sit}$ are normally distributed, independent and uncorrelated with any of the $y_i$ s. The random effects probit become:

$$P[(V_i = S = 1)] = \Pr[S > 0] = \emptyset(y_{it} \delta + \theta_i + \epsilon_{it}) \qquad (5)$$

The sectoral choices of informal non-farm employees in any of the service-oriented enterprises may be non-random, with a potential selectivity bias as a result of unobserved heterogeneity. Eq 1 as outlined above is valid when earnings from each NFHE are observed which represents a subset of the population. Thus, no selection bias is observed if this subset of the population is randomly selected. But, the decision to operate a given NFHE is often non-random—therefore requiring control for selectivity in the substantive model. Bearing this in mind, Lee (1983) further recommends a two-stage least squares to control for this. Thus, (1) becomes:

$$lnProd_{si} = X_{si}\beta_s + \lambda_{si}\delta_s + \eta_{si} \qquad (6)$$

where, $\lambda_{si} = \frac{\tau_{is\theta}(\Theta^{-1}[P_{iS}])}{P_{iS}}$ implying the selectivity term or the inverse Mill's ratioand $n_{si}$ is the random element with zero mean. Since employees are observed to be in any of the service-oriented enterprises, the earning equation is however conditional on each sector, while the inverse mill ratios are generated from the predicted probabilities from Eq (4).

Eq (6) is consistent estimate of βj if λsi is significant, indicating presence of selectivity.

Similarly, controlling for time-invariant unobservable in the earnings equation, the study further takes advantage of the panel structure of the dataset. This is called for, given that the pooled OLS of Eqs (1) and (6) may not account for such effects [42]. The FE model provides consistent estimates only on the condition that the explanatory variables are time-invariant. The time-invariant attributes in our sample such as geopolitical zones, gender of the owner, are some important determinants of productivity that should not be ignored. To ably capture these characteristics, we apply a slight modification of the fixed effect model—Between Estimator (BE) and the Random Effects (RE) model. BE is a pooled OLS estimator applied to the means of dependent and explanatory variables while controlling for the unobserved fixed

effects (see [42]. The RE model is presented as:

$$lnProd_{sit} = X_{sit}\beta_s + \rho_{sit} \tag{7}$$

where $\rho_{sit} = \alpha_{si} + \varphi_{sit}$ and $\alpha_{si}$ is the unobserved effect. Accordingly, the unobserved effects are treated as a random variable in the disturbance term. In the absence of any selectivity bias, Eq (7) yields a consistent estimate over pooled OLS if $\alpha_{si}$ is distributed independently of all $X_s$.

## 5. Empirical results

### 5.1 Choice models

We begin by analysing the probit models (pooled and random effects) as reported in Table 6 and Table A2 in S1 Appendix (see Appendix sections for the related tables). Thus, we present both the coefficients and the average marginal effects (AME—which basically clarifies the degrees of change) estimates accordingly. Table 6 reports the full model by including HBEs as an explanatory variable. This is to observe the place of HBEs in determining service-oriented choices.

The econometric results reveal sensitivity to the methodologies employed, showcasing subtle statistical differences between estimates from the pooled-probit and RE-probit approaches, as depicted in Table 6 and Table A2 in S1 Appendix for HBEs and non-HBEs. The likelihood ratio test (chibar2) was utilized to distinguish between these models, indicating highly significant unobserved heterogeneity, accounting for about 99% of the unexplained variation over time and perpetuating persistent heterogeneity in service-oriented choices due to time-invariant unobservables. Consequently, the RE-probit serves as the basis for interpretation.

It illustrates that aspects like being an HBE or not, owner's age, educational level, gender, macro-regions, electricity availability, and the presence of unpaid family members significantly determine informal wholesale, retail, and trade-oriented service enterprises overall.

Our findings are consistent with studies in other developing nations, where demographic attributes such as gender, age, and education predominantly influence wholesale and retail trade operations [43,44]. Particularly, HBEs exhibit a lower probability of engaging in wholesale retail and trade-oriented activities, roughly 3% lower than businesses specialized in consultancy and personal service ventures like beautification, tailoring, and laundering. This aligns with the idea that HBEs prioritize cost-minimization and efficient time utilization, reflecting the diverse nature of home-based work prevalent in developing regions [11]. Given that wholesale and retail businesses require public spaces to meet service demands, non-HBEs excel in such settings. Conversely, personal and consulting services might require a large pool of residential clientele, highlighting differences in output composition between these service types.

Table A2 in S1 Appendix delves into the distinct estimates among HBEs and non-HBEs for each service-oriented enterprise, acknowledging the heterogeneous nature of these enterprises and their locational attributes. It seeks to understand how these differences in observed and unobserved characteristics affect service orientation choices and subsequent productivity. Interestingly, both HBEs and non-HBEs show that older household members above 55 have a higher probability of engaging in wholesale retail and trade sectors relative to personal service and consultancy sectors. This mirrors observations across Africa, indicating that younger entrepreneurs face challenges in accessing capital or experience for business growth [7].

Furthermore, owners with post-secondary or tertiary education exhibit a stronger inclination towards informal wholesale retail businesses, especially in non-HBEs. Male business owners, relative to female owners, demonstrate a lower probability of engaging in wholesale and retail businesses, particularly pronounced in non-HBEs. This gender preference for home-

**Table 6. Determinants of service oriented household enterprises using (non)home-based as explanatory variable (pooled and RE-probit models).**

| VARIABLES | Wholesale and Retail Trade = 1 vs Consult = 0 | | | |
| --- | --- | --- | --- | --- |
| | Pooled-Probit | | RE-Probit | |
| | Coefficient | AME | Coefficient | AME |
| HBEs (non-HBE = 0) | -0.205*** | -0.069*** | -0.288** | -0.026** |
| | (0.041) | (0.014) | (0.134) | (0.013) |
| Age Group (Base < = 25) | | | | |
| 26–35 | 0.258*** | 0.096*** | 0.718*** | 0.090*** |
| | (0.068) | (0.026) | (0.242) | (0.029) |
| 36–45 | 0.448*** | 0.160*** | 1.372*** | 0.155*** |
| | (0.068) | (0.025) | (0.258) | (0.029) |
| 46–55 | 0.429*** | 0.154*** | 1.527*** | 0.166*** |
| | (0.075) | (0.027) | (0.286) | (0.031) |
| > 55 | 0.619*** | 0.211*** | 2.008*** | 0.189*** |
| | (0.077) | (0.027) | (0.308) | (0.030) |
| Education (Base = Others) | | | | |
| Primary | 0.197*** | 0.068*** | 0.588** | 0.063** |
| | (0.074) | (0.026) | (0.256) | (0.030) |
| Secondary | 0.148* | 0.052* | 0.527* | 0.058* |
| | (0.076) | (0.027) | (0.272) | (0.032) |
| Post-Secondary | 0.440*** | 0.142*** | 1.184*** | 0.102*** |
| | (0.093) | (0.030) | (0.350) | (0.033) |
| Male_owner | -0.947*** | -0.320*** | -3.746*** | -0.297*** |
| | (0.042) | (0.014) | (0.263) | (0.014) |
| Zones (Base = North-Cent) | | | | |
| North-East | 0.196** | 0.062** | 0.694* | 0.041* |
| | (0.085) | (0.026) | (0.376) | (0.024) |
| North-West | 0.245*** | 0.076*** | 0.718** | 0.042* |
| | (0.077) | (0.024) | (0.352) | (0.023) |
| South-East | 0.023 | 0.008 | 0.016 | 0.001 |
| | (0.073) | (0.024) | (0.329) | (0.028) |
| South-South | 0.159** | 0.051** | 0.324 | 0.023 |
| | (0.069) | (0.022) | (0.307) | (0.023) |
| South-West | -0.310*** | -0.112*** | -1.293*** | -0.150*** |
| | (0.068) | (0.024) | (0.321) | (0.033) |
| Rural | -0.024 | -0.008 | 0.062 | 0.006 |
| | (0.047) | (0.015) | (0.203) | (0.018) |
| HH-size | 0.013** | 0.004** | 0.028 | 0.003 |
| | (0.006) | (0.002) | (0.026) | (0.002) |
| Own_home | -0.004 | -0.001 | -0.106 | -0.009 |
| | (0.046) | (0.015) | (0.165) | (0.015) |
| Electric | -0.109** | -0.036** | -0.343** | -0.030** |
| | (0.047) | (0.015) | (0.159) | (0.013) |
| Transport | 0.014 | 0.005 | -0.028 | -0.003 |
| | (0.054) | (0.018) | (0.157) | (0.014) |
| Micro-finance | 0.021 | 0.007 | 0.169 | 0.014 |
| | (0.067) | (0.022) | (0.191) | (0.015) |
| Profit | -0.000 | -0.000 | -0.000 | -0.000 |
| | (0.000) | (0.000) | (0.000) | (0.000) |

*(Continued)*

**Table 6.** (Continued)

| VARIABLES | Wholesale and Retail Trade = 1 vs Consult = 0 | | | |
| --- | --- | --- | --- | --- |
| | Pooled-Probit | | RE-Probit | |
| | Coefficient | AME | Coefficient | AME |
| Paid_hh | 0.063 | 0.021 | 0.136 | 0.012 |
| | (0.042) | (0.014) | (0.117) | (0.010) |
| Unpaid_hh | 0.120*** | 0.040*** | 0.243** | 0.021** |
| | (0.042) | (0.013) | (0.114) | (0.010) |
| Constant | 0.452*** | | 2.838*** | |
| | (0.124) | | (0.488) | |
| Log Likelihood | -2928.36 | | -2374.14 | |
| $\hat{p}$(rho) | | | 0.954 | |
| | | | (0.003) | |
| Likelihood ratio | | | 1108 | |
| P-Value | | | 0.000 | |
| Observations | 5,394 | 5,394 | 5,394 | 5,394 |
| Number of id | | | 3,363 | |

Robust standard errors in parentheses;

***, ** and * significant at 1%, 5% and 10% respectively.

Note: AME = Average Marginal Effects.

based wholesale and retail trade might stem from the flexibility it offers in balancing household duties and labour market activities [31,45,46].

Geopolitical disparities notably influence the likelihood of operating HBEs and non-HBEs in wholesale and retail sectors. Non-HBEs show a higher probability in North (East and West) regions and a lower probability in the South-West compared to consultancy and personal service types, hinting at macro-regional variations in business locations within Nigeria. These findings resonate with earlier research highlighting commercial prominence in these geopolitical zones [2,36].

## 5.2 Productivity estimates: HBEs and non-HBEs

Table 7 and those in the Appendix (Tables A3 and A4 in S1 Appendix) present the earnings estimates based on Eqs (1), (6) and (7). To discern between the pooled OLS and RE models, the Breusch and Pegan Largrange Multiplier test (LM-Test) [47] was conducted, confirming the consistency of RE over pooled-OLS models. Table 7 encompasses the full model (employing HBEs and sector as determinants of productivity), while Tables A3 and A4 in S1 Appendix dissect the service-oriented enterprises by location (HBE or non-HBE).

Given the variations in operating wholesale and retail or consultancy and personal-oriented enterprises across sectors or locations (HBEs and non-HBEs) as highlighted in the choice models, we have accounted for selectivity bias in the earnings equations (productivity estimates). This involved incorporating the inverse Mills ratio (lambda) into Eqs (6) and (7) to mitigate potential bias beyond the related exogenous variables [41,48]. Notably, significant selectivity terms observed in certain columns (4, 6, 7, and 9) of Table 7; columns (1, 2, and 3) of Table A4 in S1 Appendix; and column (2) of Table A4 in S1 Appendix indicate possible selectivity bias in those models. Consequently, we've retained estimates where lambda is significant while also retaining Eqs (1) and (7) where lambda is insignificant. The coefficients related to selectivity terms are detailed in the Appendix (Table A5 in S1 Appendix).

**Table 7. Determinants of productivity estimates for informal HBEs and non-HBEs using (Pooled OLS, between effects and random effects models).**

| | Pooled OLS | | | RANDOM EFFECTS MODEL | | |
|---|---|---|---|---|---|---|
| | Wholesale and Retail Trade 1 | Consultancy & Personal Services 2 | Both 3 | Wholesale and Retail Trade 7 | Consultancy & Personal Services 8 | Both 9 |
| C&P (WRT = 0) | | | -0.442*** | | | -0.396*** |
| | | | (0.043) | | | (0.048) |
| HBE | -0.460*** | -0.304*** | -0.424*** | -0.430*** | -0.332*** | -0.403*** |
| | (0.048) | (0.085) | (0.041) | (0.049) | (0.079) | (0.041) |
| Age Group (Base < = 25) | | | | | | |
| 26–35 | 0.196** | 0.044 | 0.140** | 0.159* | 0.073 | 0.127* |
| | (0.078) | (0.117) | (0.066) | (0.084) | (0.112) | (0.068) |
| 36–45 | 0.377*** | 0.195 | 0.322*** | 0.323*** | 0.188 | 0.279*** |
| | (0.077) | (0.122) | (0.066) | (0.085) | (0.120) | (0.070) |
| 46–55 | 0.435*** | 0.200 | 0.365*** | 0.398*** | 0.180 | 0.321*** |
| | (0.086) | (0.140) | (0.073) | (0.094) | (0.133) | (0.077) |
| > 55 | 0.333*** | 0.126 | 0.270*** | 0.355*** | 0.125 | 0.292*** |
| | (0.088) | (0.140) | (0.075) | (0.096) | (0.142) | (0.079) |
| Education (Base = Others) | | | | | | |
| Primary | 0.014 | 0.121 | 0.044 | -0.014 | 0.182 | 0.025 |
| | (0.084) | (0.127) | (0.070) | (0.088) | (0.136) | (0.074) |
| Secondary | 0.293*** | 0.325** | 0.303*** | 0.253*** | 0.367*** | 0.272*** |
| | (0.089) | (0.138) | (0.075) | (0.092) | (0.141) | (0.078) |
| Post-Secondary | 0.667*** | 0.694*** | 0.676*** | 0.629*** | 0.762*** | 0.646*** |
| | (0.108) | (0.177) | (0.093) | (0.113) | (0.179) | (0.096) |
| Male_owner | 0.648*** | 0.627*** | 0.625*** | 0.629*** | 0.611*** | 0.615*** |
| | (0.053) | (0.082) | (0.043) | (0.059) | (0.085) | (0.049) |
| Zones (Base = North-Cent) | | | | | | |
| North-East | -0.825*** | -0.528*** | -0.757*** | -0.957*** | -0.561*** | -0.852*** |
| | (0.107) | (0.171) | (0.090) | (0.120) | (0.176) | (0.102) |
| North-West | -0.366*** | -0.538*** | -0.412*** | -0.570*** | -0.526*** | -0.566*** |
| | (0.098) | (0.136) | (0.080) | (0.120) | (0.154) | (0.101) |
| South-East | -0.294*** | 0.002 | -0.233*** | -0.301*** | -0.023 | -0.224*** |
| | (0.083) | (0.136) | (0.070) | (0.096) | (0.152) | (0.082) |
| South-South | 0.244*** | 0.267** | 0.245*** | 0.239*** | 0.316** | 0.255*** |
| | (0.080) | (0.131) | (0.068) | (0.089) | (0.143) | (0.076) |
| South-West | -0.174** | -0.238** | -0.205*** | -0.172* | -0.234* | -0.188** |
| | (0.083) | (0.116) | (0.067) | (0.093) | (0.133) | (0.077) |
| Rural | -0.058 | -0.120 | -0.061 | -0.035 | -0.148 | -0.049 |
| | (0.053) | (0.093) | (0.046) | (0.062) | (0.097) | (0.054) |
| HH-size | -0.021*** | -0.023* | -0.021*** | -0.019** | -0.018 | -0.019*** |
| | (0.007) | (0.012) | (0.006) | (0.008) | (0.013) | (0.007) |
| Electric | 0.192*** | 0.112 | 0.174*** | 0.150*** | 0.064 | 0.136*** |
| | (0.054) | (0.092) | (0.046) | (0.055) | (0.089) | (0.047) |
| Transport | 0.051 | 0.104 | 0.063 | 0.042 | 0.081 | 0.056 |
| | (0.060) | (0.087) | (0.050) | (0.059) | (0.090) | (0.049) |

(*Continued*)

**Table 7.** (Continued)

| | Pooled OLS | | | RANDOM EFFECTS MODEL | | |
|---|---|---|---|---|---|---|
| | Wholesale and Retail Trade 1 | Consultancy & Personal Services 2 | Both 3 | Wholesale and Retail Trade 7 | Consultancy & Personal Services 8 | Both 9 |
| Micro-finance | 0.028 | 0.274** | 0.093 | -0.081 | 0.300*** | 0.024 |
| | (0.081) | (0.134) | (0.070) | (0.076) | (0.111) | (0.063) |
| Turnover | 0.004*** | 0.008 | 0.004*** | 0.004*** | 0.007 | 0.004*** |
| | (0.000) | (0.007) | (0.000) | (0.001) | (0.005) | (0.001) |
| Paid_hh | -0.462*** | -0.528*** | -0.479*** | -0.301*** | -0.420*** | -0.328*** |
| | (0.049) | (0.075) | (0.041) | (0.045) | (0.069) | (0.038) |
| Inverse mills ratio | NS | NS | NS | S | NS | S |
| (lambda) | | | | | | |
| Constant | 9.021*** | 8.690*** | 9.058*** | 9.018*** | 8.649*** | 9.043*** |
| | (0.139) | (0.218) | (0.118) | (0.152) | (0.236) | (0.129) |
| Observations | 3,559 | 1,304 | 4,863 | 3,559 | 1,304 | 4,863 |
| R-squared | 0.220 | 0.238 | 0.225 | 0.217 | 0.235 | 0.223 |
| Number of id | | | | 2,300 | 940 | 3,117 |
| LM test Statistic | | | | 163.440 | 43.840 | 228.67 |
| P-Value | | | | 0.000 | 0.000 | 0.000 |

Robust standard errors in parentheses.

***, ** and * significant at 1%, 5% and 10% respectively.

C&P = Consultancy and Personal Services; WRT = Wholesale and Retail Trade; S implies that the selectivity term is significant in applied model, otherwise, NS (insignificant).

The coefficients of the RE models, observed across Table 7, Tables A3, and A4 in S1 Appendix, exhibit considerably smaller magnitudes than those of the pooled OLS. This signifies the substantial effects of time-invariant unobserved heterogeneity across service-oriented enterprises. For instance, Table 7 implies that HBEs are less productive than non-HBEs, resulting in lower value added in both service-oriented enterprises. Comparatively, productivity in the wholesale retail trade sector (-0.430) is lower than that in consultancy and personal service-oriented types (-0.332), possibly linked to [4]'s fungible allocation hypothesis. This suggests cash from sales gets reallocated into household consumption, leading to reduced earnings from businesses. Other aspects of "extended fungibility," involving time allocation between running the business and household activities, contribute to lower value added for HBEs [1,7]. Similarly, personal and consultancy-oriented enterprises consistently exhibit significantly lower productivity in all models. Hence, justifying the estimation of separate equations for HBEs and non-HBEs across service-oriented enterprises for a comprehensive analysis. To maintain brevity, inferences will primarily rely on the RE models, with the BE-model provided for robust checks.

**Productivity and informal home-based and non-homebased service oriented enterprises.** As observed from Tables A3 and A4 in S1 Appendix, owner's age, gender, educational level, geopolitical location, regional placement, enhanced electricity access, turnover, and the involvement of paid family workers serve as productivity determinants in both informal HBEs and non-HBEs. A comparative review of Tables A3 and A4 in S1 Appendix reveals a noteworthy pattern: the coefficients for post-secondary education in the wholesale retail and trade sector, initially negative in the OLS model, shift to positive in the RE models. This change

possibly reflects the influence of diverse random factors within the model, hinting at a heterogeneous impact.

Table A3 in S1 Appendix underscores the statistical significance of various labour and demographic attributes. As outlined by [7], educational attainment, gender, and regional characteristics significantly influence earnings in Africa's informal enterprises. Focusing exclusively on these variables, individuals with secondary and post-secondary education certificates demonstrate higher productivity, particularly within the consultancy and personal service-oriented enterprises, aligning with previous findings by [2,7,36] regarding the impact of human capital on productivity.

A noticeable gender earnings disparity is observed, with male earnings premium notably higher in certain sectors. In HBEs specializing in wholesale retail and trade, the male earnings premium accounts for 73.4% of the log of earnings, whereas in the personal and consultancy-oriented sectors, it constitutes 58.7%. Contrarily, within non-HBEs, male earning premiums are considerably larger in personal and consultancy-oriented work (56% of log earnings) compared to the wholesale retail and trade sector (45.3% of log earnings). This gender-based inequality aligns with studies by [7], demonstrating a larger male earning premium among nonfarm business owners in various regions, indicating the need for further empirical investigation beyond the scope of this study.

Geopolitical disparities also influence productivity, notably impacting informal HB service-oriented enterprises. Businesses within North-East and North-West geopolitical zones exhibit lower productivity compared to their counterparts in the North-Central region. Additionally, economic opportunities in the South-West favor more productive HBEs in the wholesale and trade sector compared to those in the North-Central region [36].

Infrastructure and institutional aspects also play a crucial role. Improved electricity access positively correlates with increased productivity in the wholesale and trade sector. Interestingly, within the consultancy and personal service-oriented sector, the mere presence of microfinance institutions correlates negatively with productivity. This negative association might stem from these enterprises' inability to secure loans for business growth from such institutions, consequently affecting their productive outcomes as indicated by [7].

## 6. Conclusion

The study aims to determine whether location (Home-based and Non-Home-Based) significantly impacts various service-oriented industries. It leverages three waves of the General Household Survey (GHS) panel data set (2010/2011; 2012/2013; 2014/2015) to probe the determinants influencing participation in service-oriented informal enterprises and their productivity drivers. Employing Pooled and RE-probit analyses revealed sensitive econometric results. Notably, HBEs exhibit lower probabilities of engaging in wholesale or consultancy-oriented ventures compared to non-HBEs relative to personal/consultancy-oriented types.

Further breakdowns by enterprise location (HBE versus non-HBE) and activity (wholesale, retail, trade, or consultancy/personal service-oriented) unveil heterogeneous characteristics within informal service-oriented family enterprises. It underscores the relevance of business location in shaping owners' participation and earnings in Nigeria. Noteworthy determinants encompass location (HBE or non-HBE), owner's age, education level, gender, macro-regions, electricity availability, and the presence of unpaid family members. These determinants align with studies in other developing countries, emphasizing the lower probability of HBEs engaging in wholesale compared to non-HBEs relative to consultancy-oriented sectors.

Regarding earnings (value added), HBEs are less productive than non-HBEs, potentially linked to resource allocation from businesses into family consumption. Factors impacting

productivity vary between HBEs and non-HBEs, emphasizing differential effects of business location on productivity in Nigeria's service-oriented enterprises. Geographical and demographic variables significantly influence productivity in service-oriented NFHEs, both HBEs and non-HBEs. These findings underscore the diverse impact of business location on choices and productivity among service-led family enterprises in Nigeria.

Recommendations include addressing gender disparities in participation and earnings, enhancing financial inclusion for small businesses, improving infrastructure like electricity and transport systems, and formulating policies acknowledging and supporting the informal sector to alleviate unemployment and poverty in Nigeria.

The study suggests further investigations into productivity differences between formal and informal family non-farm enterprises in Sub-Saharan Africa. It advocates exploring alternative techniques like Fuzzy Sets Qualitative Comparative Analysis (FsQCA) or Latent Growth Curve Modelling (LGCM) to derive deeper insights from longitudinal data, enhancing future research in this domain.

## Supporting information

**S1 Appendix.**
(DOCX)

**S1 Data.**
(DO)

## Author Contributions

**Conceptualization:** Ikechukwu Darlington Nwaka.

**Data curation:** Ikechukwu Darlington Nwaka, Okechukwu Lawrence Emeagwali.

**Formal analysis:** Ikechukwu Darlington Nwaka, Okechukwu Lawrence Emeagwali.

**Methodology:** Ikechukwu Darlington Nwaka.

**Supervision:** Ikechukwu Darlington Nwaka, Okechukwu Lawrence Emeagwali.

**Validation:** Okechukwu Lawrence Emeagwali.

**Writing – original draft:** Ikechukwu Darlington Nwaka, Okechukwu Lawrence Emeagwali.

**Writing – review & editing:** Okechukwu Lawrence Emeagwali.

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
