## [Decision Letter · Decision Letter 0]

3 Nov 2023

PONE-D-23-24888Participation and returns from informal service-oriented non-farm enterprises: Evidence from a survey of Nigerian householdsPLOS ONE

Dear Dr. Nwaka,

Thank you for submitting your manuscript to PLOS ONE. After careful consideration, we feel that it has merit but does not fully meet PLOS ONE’s publication criteria as it currently stands. Therefore, we invite you to submit a revised version of the manuscript that addresses the points raised during the review process.Include policy recommendations in your abstract as wellAddress any comments raised by the reviewer(s) attached herein

We look forward to receiving your revised manuscript.

Kind regards,

Stephen Esaku

Academic Editor

PLOS ONE

5. Please include your tables as part of your main manuscript and remove the individual files. Please note that supplementary tables (should remain/ be uploaded) as separate "supporting information" files

Additional Editor Comments:

Dear Author(s),

Thank you for submitting your manuscript to PLOS ONE. Kindly include policy implications of your findings in the abstract as well

Reviewers' comments:

Reviewer's Responses to Questions

**Comments to the Author**

1. Is the manuscript technically sound, and do the data support the conclusions?

Reviewer #1: Yes

2. Has the statistical analysis been performed appropriately and rigorously? 

Reviewer #1: Yes

3. Have the authors made all data underlying the findings in their manuscript fully available?

Reviewer #1: Yes

4. Is the manuscript presented in an intelligible fashion and written in standard English?

Reviewer #1: Yes

5. Review Comments to the Author

Reviewer #1: Some typos or grammar issues are highlighted within the manuscript with comment added. See attached file

The format of citations is not correct, it should be numerical and also arrange Reference in the same numerical order.

6. PLOS authors have the option to publish the peer review history of their article (what does this mean?). If published, this will include your full peer review and any attached files.

Reviewer #1: No

---

## [Author Response · Author response to Decision Letter 0]

26 Jan 2024

We are grateful for the constructive feedback, which greatly contributed to the enhancement of the manuscript and have addressed each point meticulously in the revised manuscript.

1. Policy Recommendations in Abstract: We have incorporated policy recommendations into the abstract (highlighted in red font) to offer actionable insights derived from the study's findings.

2. Addressed Reviewer Comments: We have thoroughly addressed all the comments raised by the reviewers, ensuring clarity, accuracy, and completeness throughout the manuscript. Each reviewer's concern has been carefully considered and incorporated as appropriate.

3. Separate Captions for Figures: Captions for each figure have been added as a separate section in the manuscript to comply with the journal's requirements.

4. Tables in Main Manuscript: All tables have been integrated into the main manuscript (highlighted in red font) as per the instructions. Supplementary tables have been uploaded separately as "supporting information" files.

5. Updated Reference List: We have reviewed and revised the reference list to ensure accuracy and completeness, adhering strictly to the PLOS ONE format. Retracted papers have been addressed in accordance with the journal's guidelines.

Attached is the revised manuscript, clearly marked with changes made, including the incorporated policy recommendations, updated figures' captions, and revised reference list.

---

## [Editor Report · Decision Letter 1]

31 Jan 2024

Participation and returns from informal service-oriented non-farm enterprises: Evidence from a survey of Nigerian households

PONE-D-23-24888R1

Dear Dr. Nwaka,

We’re pleased to inform you that your manuscript has been judged scientifically suitable for publication and will be formally accepted for publication once it meets all outstanding technical requirements.

Kind regards,

Stephen Esaku

Academic Editor

PLOS ONE
---

## [Editor Report · Acceptance letter]

7 Mar 2024

PONE-D-23-24888R1 

PLOS ONE

Dear Dr. Nwaka, 

I'm pleased to inform you that your manuscript has been deemed suitable for publication in PLOS ONE. Congratulations! Your manuscript is now being handed over to our production team.

Kind regards, 

on behalf of

Dr. Stephen Esaku 

Academic Editor

PLOS ONE